# Single Image Video Prediction with Auto-Regressive GANs

**DOI:** 10.3390/s22093533

**Published:** 2022-05-06

**Authors:** Jiahui Huang, Yew Ken Chia, Samson Yu, Kevin Yee, Dennis Küster, Eva G. Krumhuber, Dorien Herremans, Gemma Roig

**Affiliations:** 1Department of Electrical and Computer Engineering, University of British Columbia, Vancouver, BC V6T 1Z4, Canada; gabrie20@cs.ubc.ca; 2Information Systems Technology and Design (ISTD), Singapore University of Technology and Design, Singapore 487372, Singapore; chiayewken@gmail.com (Y.K.C.); samyubj@gmail.com (S.Y.); kevin-yee@outlook.com (K.Y.); dorien_herremans@sutd.edu.sg (D.H.); 3Cognitive Systems Lab, Department of Mathematics and Computer Science, University of Bremen, 28359 Bremen, Germany; kuester@uni-bremen.de; 4Department of Experimental Psychology, University College London, 26 Bedford Way, London WC1H 0AP, UK; 5Department of Computer Science, Goethe University Frankfurt, Robert-Meyer-Str. 11-15, 60325 Frankfurt, Germany; roig@cs.uni-frankfurt.de

**Keywords:** video prediction, autoregressive GANs, emotion generation

## Abstract

In this paper, we introduce an approach for future frames prediction based on a single input image. Our method is able to generate an entire video sequence based on the information contained in the input frame. We adopt an autoregressive approach in our generation process, i.e., the output from each time step is fed as the input to the next step. Unlike other video prediction methods that use “one shot” generation, our method is able to preserve much more details from the input image, while also capturing the critical pixel-level changes between the frames. We overcome the problem of generation quality degradation by introducing a “complementary mask” module in our architecture, and we show that this allows the model to only focus on the generation of the pixels that need to be changed, and to reuse those that should remain static from its previous frame. We empirically validate our methods against various video prediction models on the UT Dallas Dataset, and show that our approach is able to generate high quality realistic video sequences from one static input image. In addition, we also validate the robustness of our method by testing a pre-trained model on the unseen ADFES facial expression dataset. We also provide qualitative results of our model tested on a human action dataset: The Weizmann Action database.

## 1. Introduction

Recent advancements in generative neural networks have greatly improved the quality of image generation [1,2,3,4]. In the domain of video generation, however, the problem becomes much more intricate because the temporal information is introduced as another dimension. Most results produced by the existing methods are still not satisfactory to human eyes [5,6,7,8]. Our proposed approach focuses on a specific task in the domain of video generation: generating a sequence of frames based on the information provided by a single still image. We name this task “Single image video prediction”.

Many of the existing methods for video prediction take multiple frames as the input to their models, and then extrapolate the future frames by observing the trend of pixel movements within those input frames [9,10]. We view single image video prediction as a much harder task because no temporal information is provided by the input, and the model has to take into account that movements of the pixels predicted in the future frames should adhere to a timeline. We show that this ability to predict a “movement-based timeline” can be learned by our model during the training phase under certain conditions when we detail our proposed model in Section 2.1. While single image video prediction is a significantly harder task, we believe it has a wide range of applications. One example is turning static images into vivid GIFs, or making clips of different facial expressions based on the same input image. Moreover, the generated results could be used in perception studies to examine whether the animated GIFs enhance emotion perception of otherwise static images and, e.g., websites. Eventually such a tool might be used to better understand how humans may engage in a similar process of extrapolating single images in interactions. For instance, video sequences might be generated on the fly, and depending on user inputs, but in a more experimentally controlled manner than through other means, such as pre-recorded videos.

In this paper, we introduce an autoregressive approach to the single image video prediction problem. As illustrated in Figure 1, the model will constantly produce one frame at a time based on the frame generated by the previous time step, and this process is initiated and conditioned with the input image as the starter frame. This approach has several advantages: Firstly, the information is passed smoothly between the frames. Secondly, the model is able to generate videos with arbitrary lengths. Lastly, because the generation process is sequential, the resulting videos show a smooth trajectory of changes from frame to frame, thus following a continuous and consistent timeline.

However, this generation process could have a significant drawback: cumulative quality degradation. As the generation process goes on, noises and undesirable artifacts accumulate, and as a result, the generation quality suffers incrementally over time. To overcome this problem, we introduce a mechanism that we refer to as complementary masking into our model. This mechanism learns to filter out the pixels that should remain static in the generation process, and lets the model focus on the pixels that need to be changed. We show in Section 3.3 that our complementary masking mechanism is very effective when dealing with cumulative quality degradation.

We conduct extensive experiments to validate our method against other competing approaches, and show that our method is of superior quality. In addition, we demonstrate the robustness of our model by testing a pre-trained model on an unseen facial expression dataset. We also conduct experiments on an action dataset to show the generalization of our model to another video domain.

## 2. Materials and Methods

### 2.1. Proposed Video Autoregressive GAN

Here, we introduce the notation that we use in the rest of the paper, as well as the components of our proposed autoregressive generative network for single image video prediction.

#### 2.1.1. Autoregressive Generation

Our main goal is to generate a sequence of *T* video frames, denoted as $V^{\prime }:={\left\{f_{t}^{\prime }\right\}}_{t=1}^{T}$, given an input image $f_{0}\in R^{H\times W\times 3}$ as its starter frame. The generation process is autoregressive, meaning for each time step, the input to the generator network is always the output of the previous time step. The algorithm behaves slightly different in the training and testing phases. During the training phase, the ground truth frame $f_{t}$ is taken from the training video $V:={\left\{f_{t}\right\}}_{t=0}^{T}$ at time step *t*, the losses for both generator G and discriminator D networks are calculated based on the true $f_{t+1}$ and generated $f_{t+1}^{\prime }$. During the testing phase, the only input to the process is $f_{0}$, and the rest of the frames ${\left\{f_{t}^{\prime }\right\}}_{t=1}^{T}$ are all generated based on previous generations. To reduce the discrepancy between training and testing, we propose to perform scheduled sampling (SS) [11] in the later epochs of the training phase. After some number of initial guiding epochs $E_{g}$, the model randomly switches its input between the ground truth current frame $f_{t}$ and the synthesised current frame $f_{t}^{\prime }$ produced by the previous time step, with a sampling rate *r*. In this way, we can increase the model’s exposure to its own generations, and bridge the behavioral gap between training and testing phases. The process is summarized in the following algorithm.

#### 2.1.2. The Generator Network

As discussed in the introduction, quality degradation poses a substantial problem in a sequential autoregressive generation process, because noises and undesirable artifacts will accumulate as the generation process goes on. Inspired by [5], we adopt a two-stream architecture to form a complementary masking mechanism to avoid this problem. The formulation of our output is as follows:(1)$$G\left(f_{t}\right)=m\left(f_{t}\right)⊙d\left(f_{t}\right)+\left[1-m\left(f_{t}\right)\right]⊙f_{t},$$
where G represents the generator component of the GAN model, m(·) is a function that outputs a soft mask with values between 0 and 1 $\left(0\leq m\left(f_{t}\right)\leq 1\right)$, d(·) is a function whose output can be viewed as a difference map between $f_{t}$ and $f_{t+1}$, and ⊙ is element-wise multiplication. The intuition behind Equation (Equation 1) is that when composing the next frame, some pixels should remain static compared to the current frame, which we call static pixels; while the rest of the pixels should be changed to follow the motion of the subject in the video, which we call variable pixels. Our formulation enforces a complementary relation such that the mask for static pixels and the mask for variable pixels sum up to 1. This simulates the formation of a complete frame, by adding both static and variable pixels together. We illustrate the autoregressive generation with the complementary mask in Figure 2a,b, respectively. In our ablation studies, we show that the complementary mask module helps the model to focus on the variable pixels, which in turn helps to reduce quality degradation. In our setup, we use a ResNet [12] based architecture for m(·), and a U-Net [13] based architecture for g(·).

#### 2.1.3. The Discriminator Network

We use two discriminators in our autoregressive GAN architecture, the global and the local discriminator. The global discriminator $D_{g}$ overviews the entire frame and distinguishes between the ground truth next frame and its counterpart’s generation.
(2)$$D_{g}\left(f_{t},f_{t+1}\vee f_{t+1}^{\prime }\right)=real\vee fake.$$

The local discriminator $D_{l}$ puts more attention to the variable pixels. Here, we reuse the mask generated in the generator to eliminate static pixels: by subtracting $f_{t}$ on both sides of Equation (Equation 1), we get:(3)$$f_{var,t+1}^{\prime }=m\left(f_{t}\right)⊙\left(d\left(f_{t}\right)-f_{t}\right).$$

We also subtract $f_{t}$ from the real $f_{t+1}$ to get the ground truth variable pixels:(4)$$f_{var,t+1}=f_{t+1}-f_{t}.$$

Finally, our second discriminator $D_{f}$ can be formulated as:(5)$$D_{l}\left(f_{t},f_{var,t+1}\vee f_{var,t+1}^{\prime }\right)=real\vee fake.$$

We illustrate the discriminator network in Figure 2c. Unlike other methods [14,15] that employ a predefined region (mouth region cropping, eye region cropping, etc.) for the local discriminator, we can benefit from the mask learned by the generator and automatically crop out the regions of interest. Thus, we argue that our design for the local discriminator is more flexible and generalizable to different domains and datasets, as we show in the experimental section. In our setup, we use Patch-GAN [2] for both our global and local discriminators.

#### 2.1.4. Joint Objective Function

We propose an objective function that consist of the sum of an adversarial objective function, a reconstruction loss and the mask sparsity loss.

Since we have two discriminators in our GAN architecture, the adversarial objective can be written as:
(6)$$\begin{array}{ccc} L_{GAN}\left(G,D_{g},D_{l}\right) & = & E_{f_{t+1}}\left[\log D_{g}\left(f_{t},f_{t+1}\right)\right]+\\ & & E_{f_{t+1}}\left[\log D_{l}\left(f_{t},f_{t+1}\right)\right]+\\ & & E_{f_{t}}[\log\left(1-D_{g}\left(f_{t},G\left(f_{t}\right)\right)\right]+\\ & & E_{f_{t}}[\log\left(1-D_{l}\left(f_{t},G\left(f_{t}\right)\right)\right], \end{array}$$
where G tries to minimize this objective against its adversaries $D_{g}$ and $D_{l}$ that try to maximize it:(7)$$G^{*}=\arg\;\;\min_{G}\max_{D_{g},D_{l}}L_{GAN}\left(G,D_{g},D_{l}\right).$$

We also add a reconstruction loss because it is proven to be effective for improving the quality of the generated samples with GANs. Here, we combine the traditional L1 distance with our GAN objective. While the discriminators’ task remains unchanged, the generator should produce outputs that are as close to the ground truth as possible in an L1 sense:(8)$$L_{L1}\left(G\right)=E_{f_{t}}{∥f_{t+1}-G\left(f_{t}\right)∥}_{1}.$$

Following [5], we found that adding a sparsity prior on the mask can encourage the generator to reuse as many pixels from its input as possible, thus reducing noise accumulation during the generation process:(9)$$L_{MS}\left(G\right)=\lambda {∥m\left(f_{t}\right)∥}_{1},$$
where λ here can be interpreted as the strength of this prior. A larger λ will encourage the network to reuse more pixels from its input. We set λ=0.2 in our experiments. Similar to the reconstruction loss, the mask sparsity loss only affects the generator.

To sum up, our final objective to train the network can be written as:(10)$$G^{*}=\arg\;\;\min_{G}\max_{D_{g},D_{l}}\left[L_{GAN}\left(G,D_{g},D_{l}\right)+L_{L1}\left(G\right)+L_{MS}\left(G\right)\right].$$

### 2.2. Implementation

As discussed above, we have two streams in our generator G: a function m(·) that outputs a mask, and a function d(·) that generates the difference map. For m(·), we use a ResNet [12] based architecture with two downsampling/upsampling layers scattered between 9 residual blocks, and we use the tanh function to rescale the output to [0,1]. For d(·), we employ an 8-layer U-Net [13] with skip connections to enable the flow of both low-level and high-level features. We use the PatchGAN [2] discriminator with 70×70 patch size for both the global and local discriminators. We set the video batch size to 1 and sample 10 frames from each individual clip for training. We use the Adam [16] optimizer and set the learning rate to 0.0002, the learning rate linearly decays to 0 starting from the half of the total epochs. The scheduled sampling process starts at the half of the total epochs in the training. We set our initial sampling rate to 0.1 and increase this by 0.1 for every 20 epochs. All input frames are resized to 256×256 and the values normalized to [−1,1]. We train separate models for each emotion, and we do not have classifiers for emotion classification. The same approach is used for human actions, in which we have one model per action. We will make the code publicly available upon acceptance of the paper.

### 2.3. Datasets

#### 2.3.1. Facial Expression Generation

We require each training sample to contain a single facial expression that begins with a neutral expression, and reaches a single apex of expression intensity. We use the UT Dallas dataset [17] as our base dataset. Considering that this dataset holds a different number of samples for different facial expressions, we selected 4 expression classes with the largest number of videos: Happiness (316 videos), Disgust (254 videos), Surprise (192 videos), and Sadness (60 videos). We trim these videos into clips between 1–2 s to remove any idling frames. Following this, we sample 10 frames for each clip and resize them to 256×256 pixels. For each emotion class, we take 20 different subjects for the testing split, and the remainder for the training split.

#### 2.3.2. Human Actions Generation

We use the Weizmann Action database [18], and selected 4 different actions (“one-hand wave”, “two-hands wave”, “skip” and “jump”) to train and test our model. Each action category contains 9 videos of different performers, we trim the videos to make sure there is no repetitions. We randomly select 7 subjects for training and 2 subjects for testing. All frames are resized to 64×64 in size. Due to the small sample size, we did not conduct a quantitative evaluation on this dataset. Instead, we provide visual results in Figure 3, to showcase the generalizability of our method to different video domains.

## 3. Results

In this section, we provide detailed analysis and results of our model on different perspectives, including an ablation study on the effects of different components of our model design, and comparison to other methods in the literature.

### 3.1. Quantitative Results

For quantitative comparisons, we use Frechet Inception Distance (FID, a lower value indicates better quality) [19] and the Inception Score (IS, a higher value indicates better quality) [20] to measure perceptual quality and diversity. On Table 1, we compare our method against two other temporal-based video generation models: ConvLSTM [6] and the flow-grounded spatial-temporal (FGST) method [7], with the ground truth frames results (first row) as reference. We also report results generated by non-temporal-based method AffineGAN [14] for comparison. All models are trained and tested on the “Happiness” class in the UT Dallas dataset. We observe that our method is considerably better than the other two temporal-based models on both evaluation metrics, and its performance is comparable to that of the AffineGAN. Notice that the AffineGAN model is unaware of the temporal information and requires additional guidance (the expression intensity score) in the generation process. The model has to scatter the expression intensity along the timeline, and then generate each frame independently to produce the final clip, thus, it should be categorized as an image-translation task, which is an easier task compared to ours.

### 3.2. Qualitative Results

Figure 4 shows the visual comparisons of our method and three other temporal-based video generation models: ConvLSTM, FGST (temporal-based) and AffineGAN (non-temporal-based), with the ground truth (GT) frames in the first row. All models are trained on the “Happiness” class with the same number of epochs, and we show our results on the test set. As shown in the figure, among the temporal-based methods, our model is able to produce what appears to be the most realistic frames with consistent continuity. We also compare our results with the non-temporal-based method AffineGAN in the last row. As discussed above, the generation process in the AffineGAN setup should be considered as an easier task compared to ours. Nevertheless, we can still observe that the results generated by our model are comparable to the AffineGAN results in both realism and continuity.

Figure 5 shows more results generated by our model trained on different emotions. Our model is able to learn some generalised traits for different emotions, for example, pouting and frowning in the “Disgust” category, lifting eyebrows in “Surprise”, and the crying face in “Sadness”. As we did not introduce any prior to the model during the training phase, all of these traits were summarised and learnt by the model itself by observing the training data.

In Figure 6, we demonstrate the robustness of our model by testing it on an unseen dataset, the Amsterdam Dynamic Facial Expression Set (ADFES) [21]. The model tested is trained on the “Happiness” class on the UT Dallas dataset. We observe that our model is able to produce frames with consistent quality and smoothness, even though the background, illumination and video setup is different from the training dataset (UT Dallas).

Figure 3 shows the results of our model trained and tested on the Weizmann Action database. We show the results generated by our model on four different action classes: “one hand wave”, “tow hands wave”, “skip” and “jump”. Our model is able to predict both the limb and body movements for the performers on the test set. This demonstrates the potential of our model for generating videos that are not limited to the facial expression domain.

### 3.3. Ablation Studies

#### 3.3.1. Complementary Mask

The complementary mask mechanism is the key component of our generator network, since it separates the static pixels and variable pixels such that the model will learn to reuse some of the pixels from the previous time step. In this way, the quality degradation problem in the autoregressive generation process appears to be greatly suppressed. Figure 7a shows the example results generated by our model with/without the complementary mask mechanism, illustrating that without the mask, the model suffers from quality degradation when generating later frames. On Table 2, numbers also suggest that implementing the complementary mask mechanism improves the generation quality of our model.

#### 3.3.2. Two Discriminators

In addition to the global discriminator in our framework, we employ another local discriminator that puts more attention to the variable pixels filtered by the mask mechanism in the generator network. It helps refine the details of the generated image, especially for the regions that are activity involved in the expression. As illustrated in Figure 7b, without any prior knowledge, the model learned to mask out the eye and mouth regions on the subject’s face, and the local discriminator pressured the generator to refine. Unlike previous methods [14,15], this entire process is fully automatic. Table 2 shows that having two discriminators in our network improves both the FID and the Inception Score.

##### Scheduled Sampling (SS)

The nature of autoregressive generation leads to discrepant behaviours during the training and the testing phases. If we only use ground truth next-frames during the training phase, the model will lack the ability to adjust itself to deal with generated frames. Figure 7c shows the effectiveness of scheduled sampling (SS) during the training process. We observe that applying SS during training results in an improvement of the vividness of the generated sequence at test time(the expression generated is more intense). Table 2 shows that performing SS slightly lowers the Inception Score, but it helps reduce the FID score. This is reasonable because the Inception Score only takes measurements on individual images, while FID measures the perceptual distance between the predicted and the target image. The number suggests that performing SS helps the model generate images that are closer to the ground truth.

### 3.4. Mask Visualization

In Figure 8, we visualize the masks produced by the generator of our architecture. Recall that the mask has 3 channels (RGB) within the range of [0,1], each channel will be multiplied with a foreground difference map, and its complement will be multiplied with the previous frame. All three channels are stacked and rescaled to a range of [0,255] for visualization. As a result, higher values correspond to the foreground change (variable pixels), and lower values correspond to the background (static pixels).

We take three different subjects in the “Happiness” category for illustration. The masks shown in the figure are sampled from the first, fourth and seventh timestep in the generation process. Interestingly, our model learnt to crop out facial regions that are “important” in facial expressions generations, such as the mouth, eyes and cheeks. This demonstrates the model’s ability to automatically detect regions of interest that may help with the generation process.

## 4. Discussion

Here, we first include previous work, including methods in the literature that are related to our approach. These are methods that do image-to-image translation and temporal based image prediction, both connected to video prediction from a single image. We have also included methods that tackle the problem of semantic foreground and background distinction, which are related to the usage of the mask in our system to avoid degradation of the predicted face over time. Then, we discuss how our proposed method is framed in relation to those methods from the literature, and our specific contributions with respect to them.

### 4.1. Previous Work

#### 4.1.1. Image-to-Image Translation

Image-to-image translation has been an active area of computer vision in recent years [22,23]. It is the task of generating a new image from another input image. The mapping between the input and output image from different domains is learned, and can then be used for applications style transfer [3,4], photo enhancement [24,25] and future state prediction [26]. Generative adversarial networks (GANs) [1] are the backbone of most recent work in this area due to their powerful capability to generate sharp images. One popular variant is the conditional GAN [27], where the model generates images with characteristics from specific class labels, rather than generic samples from an unknown noise distribution. Our proposed method follows a similar architecture as proposed in the Pix2Pix model [2]. In [2], the conditional GAN has a generic loss function that allows it to learn effective mappings between input and output images from diverse domains. Like [2], our generator has a “U-Net” structure [13] with skip-connections [12], which helps preserve low-level information that is shared between input and output images. Similarly to [2], our discriminator judges whether each N×N patch in an image is more likely to be real or fake, and outputs the average probability. This helps to reduce parameters and training time.

#### 4.1.2. Semantic Foreground-Background Distinction

For better video generation, some methods have also distinguished the foreground information from the background information. For instance, [5] explicitly models the background and foreground as two different parts. This is motivated by the observation that the background is relatively static in most videos, and helps the model to learn the motion pattern of objects. This approach of semantic distinction is also useful for video understanding applications, such as in [28], where salient movements are detected by taking into account the temporal relationships between pixels of subsequent frames. In another approach [29], they used the Mask R-CNN [30] instead of object proposals to model scene transition by segmenting the region of interest from the background.

#### 4.1.3. Temporal-based Video Prediction

Videos are composed of frames that are temporally related and that follow a sequential order. Hence, most of the previous approaches for video prediction use recurrent neural networks [8,9,31] or 3D convolutional neural networks [5,7,10] to capture the temporal relationships between the frames. For example, in Villegas et al. [9], a convolutional LSTM encoder-decoder architecture was used to predict the next frames conditioned on previous frames up to a fixed number of time steps. In [10], the authors use a spatial-temporal architecture with 3D convolution layers to capture an optical flow map which is used to generate future frames from the first frame.

Amazing results have been achieved in the domain of facial expression generation. In one such approach [14,15], an anchor image, as well as a facial expression intensity score as guidance were used to generate corresponding frames for an expression. However, this type of approach requires additional information such as facial landmark cropping [14], or AU annotation [15]. Furthermore, note that these kinds of information are very specific, and limited to the domain of facial images. In addition, these methods can only generate videos with pre-defined lengths, depending on how much they want to scatter their intensity score.

## 5. Advances of Our Approach in Relation to Previous Work

In contrast to the aforementioned previous works for generating a video sequence based on a single input image, our model is able to generate frames of an arbitrary length, which provides flexibility in generating the desired intensity of a given target expression, possibly beyond the intensity of expressions exhibited in the training data. We tackle the problem by proposing an autoregressive approach during the generation process, which uses the output generated in one time step, as input to the next step to ensure smooth trajectories of the generated video. To ensure that the quality of the generated frames is not degraded along the generative process, we introduce a complementary mask to avoid noise and artifacts accumulating over time. We reported the robustness of the model by using images from an unseen facial expression dataset. Furthermore, given that our model does not require any guidance in the generation process, this allows us to go beyond the facial expression domain with respect to testing our method on other types of videos, as demonstrated in the experiments on the Weizmann Action database, which show the robustness of the proposed method.

While generating videos from single images is a very hard task and immediate use of the method might not be possible depending on the quality needed for a given application, such as high quality animation for any expression, we believe that our method poses an advancement in the quality of the generated results with respect to previous work, and it poses a new framework in which more improvements can be potentially incorporated.

## 6. Conclusions

In this paper, we propose a solution to the single image animation problem based on an autoregressive approach. By constantly re-feeding the output of the model to itself, the model is able to generate videos with arbitrary lengths. The autoregressive approach has two distinctive advantages: (1) It helps preserve the details and the context from the starter frame. (2) It enforces the generation results to be sequential, thus following a continuous timeline. We use the complementary mask mechanism to suppress the quality degradation effect caused by the autoregressive process. We propose a two-discriminator setup that reuses the mask produced in the generator to refine details in critical regions, since this process is automatic, no extra information such as region of interest cropping is needed. We conduct extensive experiments on a facial expression dataset, and demonstrate our model is able to generate realistic and continuous facial expression videos with only one starter image and no other extra information such as expression intensity guidance. We also show the potential of our model for video generation in other domains by testing it on a human action dataset. For future work, one could explore if expanding the input of the generative model to capture all previously generated frames improves further the quality of the results.

## Figures and Tables

**Figure 1 sensors-22-03533-f001:**
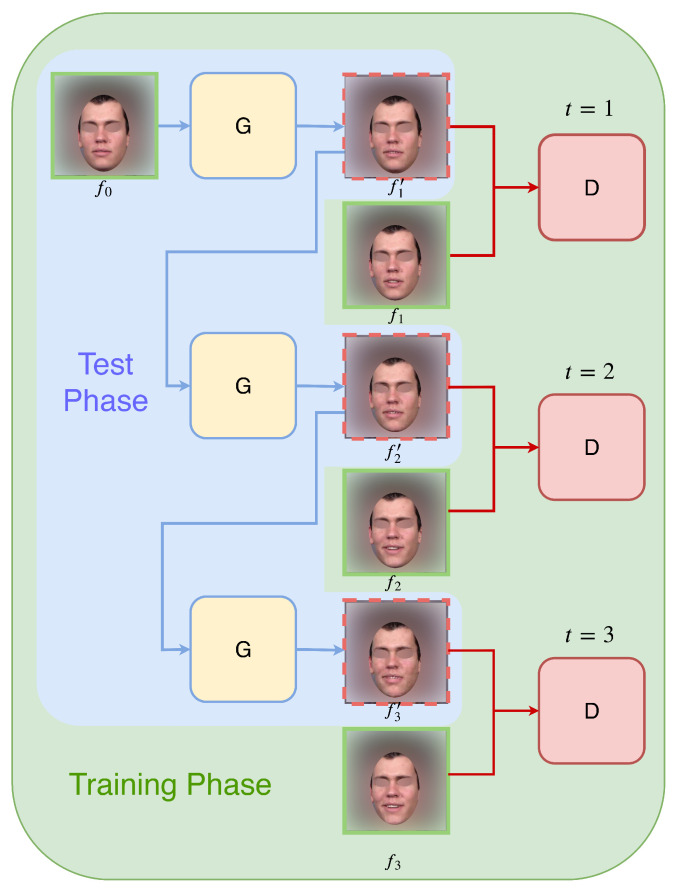
Autoregressive generation. G denotes the generator network, and D denotes the discriminator network. The model generates one frame at a time, the input to the model is always the output from the previous time step.

**Figure 2 sensors-22-03533-f002:**
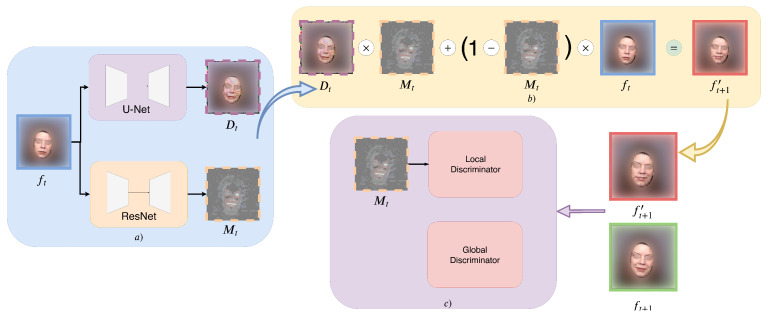
Architecture Overview. (**a**) is the generator, it takes a frame $f_{t}$ as input, and produces a difference map $D_{t}$ and a mask $M_{t}$. (**b**) is the complementary masking module that uses the mask $M_{t}$ to merge the input frame $f_{t}$ and the difference map $D_{t}$ into the predicted next frame $f_{t+1}^{\prime }$. (**c**) is the discriminator network.

**Figure 3 sensors-22-03533-f003:**
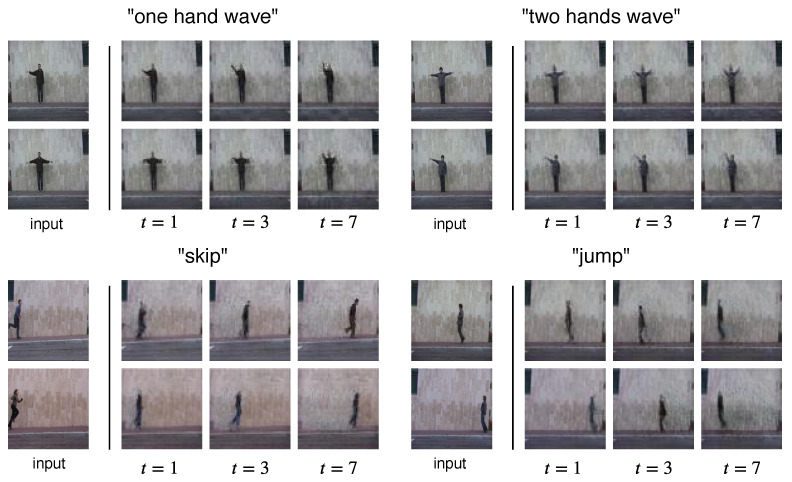
Results on the Human Action dataset.

**Figure 4 sensors-22-03533-f004:**
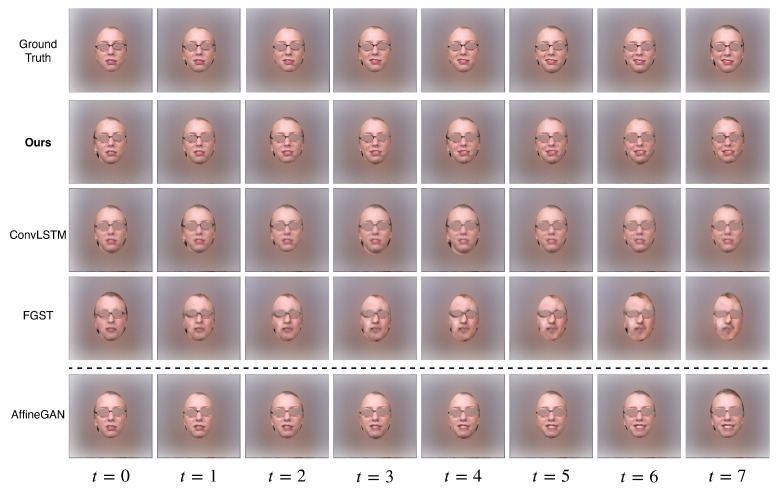
Qualitative comparison with different models for the “Happiness” class on the test set. The first row are the ground truth frames, the following rows are frames generated using different models. We also compare our results with the AffineGAN, which is not a temporal-based model. Our results are clearly better than ConvLSTM and FGST, and are comparable to results produced by AffineGAN.

**Figure 5 sensors-22-03533-f005:**
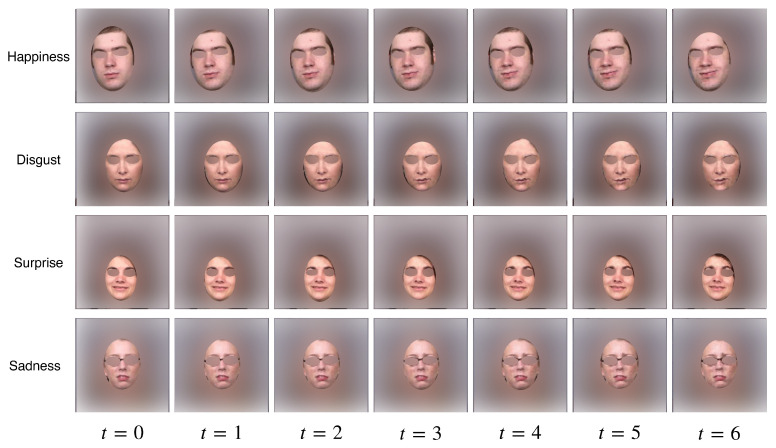
Different Emotions. We train our model on four different emotions on the UT Dallas dataset: Happiness, Disgust, Surprise and Sadness. Here, we show the results from the test set.

**Figure 6 sensors-22-03533-f006:**
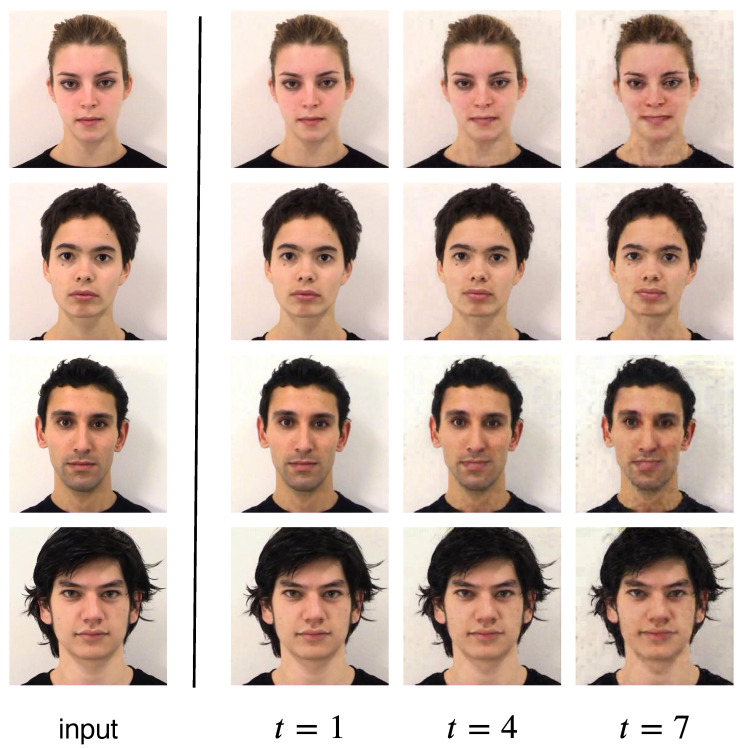
Results on the unseen ADFES dataset.

**Figure 7 sensors-22-03533-f007:**
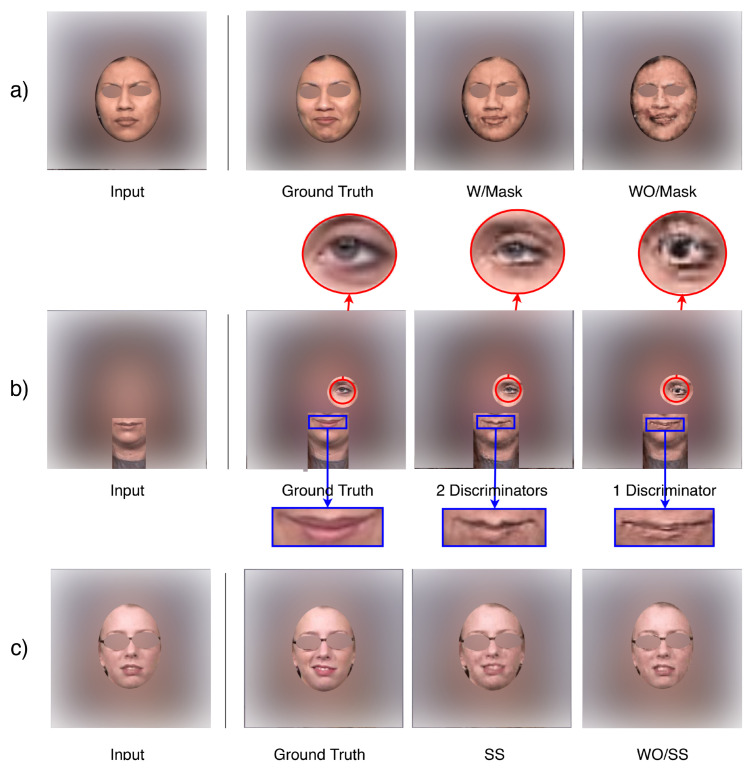
Ablation Studies. (**a**) Complementary Masking. (**b**) Two discriminators vs. one discriminator. (**c**) Scheduled Sampling (SS).

**Figure 8 sensors-22-03533-f008:**
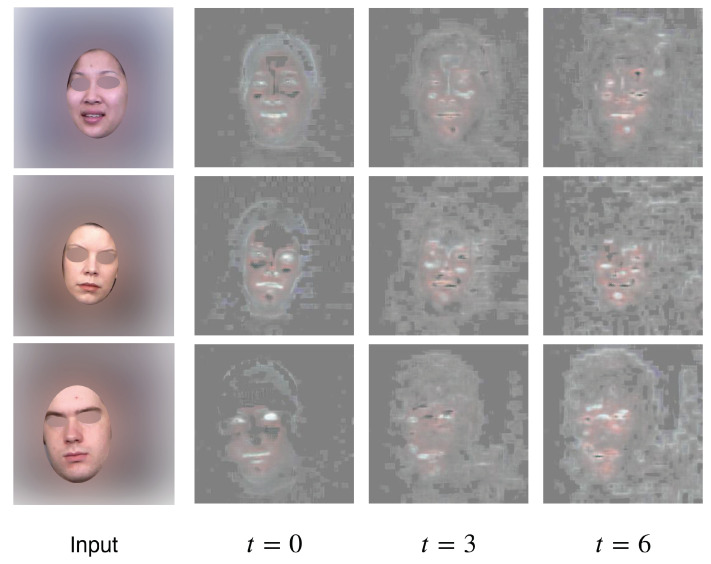
Mask Visualization. Without any manual annotation, our model learns to crop out some facial landmarks that are important for expression generation.

**Table 1 sensors-22-03533-t001:** Quantitative comparison with ConvLSTM, FGST (temporal-based) and AffineGAN (non-temporal based). We use Frechet Inception Distance (FID) and Inception Score (IS) as evaluation metrics. We report the results on the test set of the “Happiness” class.

Model	FID	IS
Ground Truth	0	2.124
ConvLSTM	95.468	1.635
FGST	116.994	1.829
Ours	33.616	2.012
AffineGAN	15.869	2.111

**Table 2 sensors-22-03533-t002:** Quantitative results for the ablation studies. All the models are trained and tested on the UT Dallas dataset, “Happiness” class.

Model	FID	IS
Ours w/o Mask	49.759	1.942
Ours w/o two discriminators	33.678	1.891
Ours w/o SS	33.821	2.024
Ours	33.616	2.012

## Data Availability

Not applicable.

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
