# Peer review of "Single Image Video Prediction with Auto-Regressive GANs"

_sensors, 2022, doi:10.3390/s22093533_

Round 1

Reviewer 1 Report

The paper propose a solution to the single image animation problem based on an autoregressive approach. It is a topic of interest to the researchers in the related areas. For the reader, however, a number of points need clarifying and certain statements require further justification. My detailed comments are as follows:

(1)What is the research significance of single-image video prediction itself?What are the practical applications?Please add this explanation in the article.

(2)Will the image requirements of the test dataset affect the performance of the system?Does the problem of small sample size guarantee the accuracy of the system in practical application?Can the authors find other data to verify the accuracy of the results?

(3)In the fourth part, the author talks about comparing and linking the research methods with other literature. Please briefly explain the specific advantages of the method.

(4)It is noteworthy that your paper needs to be carefully edited in the format.Also, use the subheading to distinguish the chapters, please use the formula editor to write the formula.

Author Response

Here we address the questions and comments raised by the reviewers:

Reviewer 1:

  • What is the research significance of single-image video prediction itself? What are the practical applications?Please add this explanation in the article.

We emphasized research significance and added more practical applications in the introduction, lines 28-36. Here is the new text for reference:

“While single image video prediction is a significantly harder task, we believe it has a wide range of applications. One example is turning static images into vivid GIFs, or making clips of different facial expressions based on the same input image. Moreover, the generated results could be used in perception studies to examine whether the animated GIFs enhance emotion perception of otherwise static images and, e.g., websites. Eventually such a tool might be used to better understand how humans may engage in a similar process of extrapolating single images in interactions. For instance, video sequences might be generated on the fly, and depending on user inputs, but in a more experimentally controlled manner than through other means, such as pre-recorded videos.”

  • Will the image requirements of the test dataset affect the performance of the system? Does the problem of small sample size guarantee the accuracy of the system in practical application? Can the authors find other data to verify the accuracy of the results?

For generating videos at test time, the requirements of the input image are that they contain a face, in case of generating a video of a facial expression, or a standing person, in case of generating an action video. To validate the robustness of our model and generalizability to an unseen dataset, we conducted experiments using the trained models on one facial expression dataset, and tested them on another facial expression dataset. This demonstrates that the complimentary mask introduces robustness to changes in the background and that our approach is able to generalize to different scenarios. To further test our method in another application different from facial expression generation, we report an experiment on a human action dataset.

  • In the fourth part, the author talks about comparing and linking the research methods with other literature. Please briefly explain the specific advantages of the method.

Thanks for the suggestion. We extended the discussion by adding in the manuscript the following paragraph in lines 262-274:

“In contrast to the aforementioned previous works for generating a video sequence based on a single input image, our model is able to generate frames of an arbitrary length, which provides flexibility in generating the desired intensity of a given target expression, possibly beyond the intensity of expressions exhibited in the training data. We tackle the problem by proposing an autoregressive approach during the generation process, which uses the output generated in one time step, as input to the next step to ensure  smooth trajectories of the generated video.  To ensure that the quality of the generated frames is not degraded along the generative process, we introduce a complementary mask to avoid noise and artifacts accumulating over time. We reported the robustness of the model by using images from an unseen facial expression dataset. Furthermore, given that our model does not require any guidance in the generation process, this allows us to go beyond the facial expression domain with respect to testing our method on other types of videos, as demonstrated in the experiments on the Weizmann Action database, which show the robustness of the proposed method.”

  • It is noteworthy that your paper needs to be carefully edited in the format. Also, use the subheading to distinguish the chapters, please use the formula editor to write the formula.

Thanks. We have addressed the formatting issues.

Due to copyright issues with UT Dallas which doesn’t allow for the publication of certain images, we would like to note that we have masked the faces of the UT Dallas Dataset to hide the identity of the appearing persons.

We will be glad to reply to further questions. We are looking forward to hearing from you.

Sincerely,

The authors

(Jiahui Huang, Yew Ken Chia, Samson Yu, Kevin Yee, Dennis Küster, Eva G. Krumhuber, Dorien Herremans, Gemma Roig)

Reviewer 2 Report

This paper aims to generate an entire video sequence based on a single input image. Unlike other video prediction methods that use “one shot” generation, this paper adopts an autoregressive approach in the generation process, which will reuse the output from each time step to the next step as the input image. This approach has several advantages: Firstly, the information is passed smoothly between the frames. Secondly, the model can generate videos with arbitrary lengths. Lastly, because the generation process is sequential, the resulting videos show a smooth trajectory of changes from frame to frame, thus following a continuous and consistent timeline.

However, this generation process could have a significant drawback: cumulative quality degradation. As the generation process goes on, noises and undesirable artifacts accumulate, and as a result, the generation quality suffers incrementally over time. To overcome the problem of generation quality degradation, this paper introduces a “complementary mask” module in the architecture, and this allows the model to only focus on the generation of the pixels that need to be changed.

Experiments on UT Dallas Dataset, unseen ADFES facial expression dataset and the Weizmann Action database show the robustness of the proposed method.

While these issues need to be addressed:

  1. From Fig 3 and 6, I did not find much movement along the time axis, is this a drawback of this approach?
  2. As “complementary mask” is introduced, the generated image will be composed by 2 images. Will this limit the movement of generated image?
  3. Figure 2 has not been indicated in the content.

Author Response

Here we address the questions and comments raised by the reviewers:

Reviewer 2:

  • From Fig 3 and 6, I did not find much movement along the time axis, is this a drawback of this approach?

The changes in the facial expression can be sometimes subtle, but the main characteristics are picked by the model, such as raising the eyebrows for surprise, and the smile for the happy emotion. Also, when looking at the images generated for the action dataset (Figure 5 in the manuscript), for which the movement can be more evident in actions such as walking, we can see that there is a substantial change at the pixel level, indicating that our method is able to generate such transitions over time.

  • As “complementary mask” is introduced, the generated image will be composed by 2 images. Will this limit the movement of generated image?

When computing and applying the complementary mask, it helps the model to focus on the variable pixels. The assumption is that in the next frame, some pixels should remain static compared to the current frame; while the rest of the pixels should be changed to follow the motion of the video. We don’t see this as a real limitation, as it is plausible to assume smoothness in the transition between consecutive frames.

  • Figure 2 has not been indicated in the content.

Thanks for noting this. We now reference it in lines 84 and 89.

Due to copyright issues with UT Dallas which doesn’t allow for the publication of certain images, we would like to note that we have masked the faces of the UT Dallas Dataset to hide the identity of the appearing persons.

We will be glad to reply to further questions. We are looking forward to hearing from you.

Sincerely,

The authors

(Jiahui Huang, Yew Ken Chia, Samson Yu, Kevin Yee, Dennis Küster, Eva G. Krumhuber, Dorien Herremans, Gemma Roig)